# The Role of Trust in Disaster Risk Reduction: A Critical Review

**DOI:** 10.3390/ijerph21010029

**Published:** 2023-12-24

**Authors:** Rubinia Celeste Bonfanti, Benedetta Oberti, Elisa Ravazzoli, Anna Rinaldi, Stefano Ruggieri, Adriano Schimmenti

**Affiliations:** 1Faculty of Human and Social Sciences, Kore University of Enna, 94100 Enna, Italy; rubiniaceleste.bonfanti@unikore.it (R.C.B.); stefano.ruggieri@unikore.it (S.R.); 2Center for Climate Change and Transformation, Eurac Research, Viale Druso 1, 39100 Bolzano, Italy; benedetta.oberti@eurac.edu (B.O.); elisa.ravazzoli@eurac.edu (E.R.); 3Institute for Regional Development, Eurac Research, Viale Druso 1, 39100 Bolzano, Italy; 4Department of Economics, University of Bari “Aldo Moro”, Largo Abbazia S. Scolastica, 70124 Bari, Italy; anna.rinaldi@uniba.it

**Keywords:** trust, community, natural disasters, disaster risk reduction

## Abstract

In recent years, there has been a growing interest in the concept of trust within the domain of natural disaster management. Trust can be defined as a state of vulnerability where one party relies on another party with the expectation that the latter will carry out entrusted responsibilities without exploiting this inherent vulnerability. This comprehensive literature review is dedicated to the examination of research concerning community and institutional trust in the field of disaster risk reduction (DRR). Particular emphasis is placed on elucidating the influence of trust throughout the distinct phases of natural disaster management, namely prevention, preparedness, response, and recovery. The critical examination of the pertinent body of the literature demonstrates that trust plays a central role across the different phases of DRR, being positively associated with effective community responses and resilience. Hence, it becomes imperative to actively foster the development of trust at both institutional and community levels within the realm of DRR. This endeavor is essential for adequately preparing communities to confront natural disasters, crafting effective protocols to enhance community responsiveness and mitigate adverse consequences, and advancing strategies for successful reconstruction and recovery.

## 1. Introduction

In recent years, there has been an increasing scholarly interest in the examination of trust within the domain of disaster risk reduction (DRR) [1,2,3,4]. Researchers hailing from diverse disciplines, including economics, law, psychology, and social sciences, have collectively recognized trust as a critical factor impacting the level of community response when confronted with natural disasters.

Trust can be broadly defined as “a psychological state comprising the intention to accept vulnerability based upon positive expectations of the intentions or behavior of another” [5]. The concept of trust assumes a fundamental role in comprehending the dynamics of social interactions among individuals and groups. Trust necessitates the willing acceptance of vulnerability by the trusting party when engaging with a trusted entity—whether it be an individual, a group, or an institution—without the presence of immediate guarantees or assurances regarding the motivations and behaviors of the trusted party [6,7].

Consequently, trust transcends individual boundaries and maintains a close association with the concept of community [4,8]. As elucidated by Walker and colleagues [9], trust is essential for forming and maintaining connections between individuals who may not interact otherwise. In this context, trust can be considered a relational construct that defines the quality of interpersonal communication between individuals or groups. It enables individuals to establish relationships within their communities and with institutions, and it facilitates the exchange of support and assistance [10]. Specifically, community trust can be characterized as a fundamental attribute of communities that implies the presence of confidence in participation and the belief that a community possesses the capacity to resolve disputes and engage in collectively accepted public endeavors [11]. 

Notably, trust exhibits the capacity to mitigate the adverse impacts of psychological distress experienced by marginalized or economically disadvantaged groups [12]. Conversely, numerous findings indicate that the community’s resilience levels are negatively affected by insufficient levels of trust and unsupportive governance structures, hindering access to resources and inclusion in DRR processes. This, in turn, perpetuates a self-reinforcing cycle that erodes trust [13]. 

Furthermore, it has been observed that trust in institutions, environmental groups, and scientists positively correlates with the adoption of appropriate behaviors in DRR. These behaviors include individual-level preparatory actions, e.g., the acquisition of insurance and the endorsement of adaptation policies [8,13]. For this reason, in recent years, there has also been a growing interest in the examination of trust concerning government and institutional entities. The notion of institutional trust hinges on the belief in the capability of institutions to effectively manage a diverse array of risks and social challenges [14]. This concept is frequently associated with the expectation that institutions will implement policies that are advantageous and successful for the well-being of citizens and is regarded as a significant gauge of broad-based political endorsement [9].

Institutional trust assumes particular significance when risks and their potentially negative impact on citizens emanate primarily from sources beyond individual control, as well as when the institution is entrusted with the responsibility of forecasting and addressing damages that are largely outside the control of citizens—such as in situations involving natural hazards, where citizens cannot predict the entity of damage and have little control over recovery processes. The significance of trust in reducing the complexities and costs of risks lies in fostering a robust sense of concern, solidarity, and active engagement within the community. This, in turn, may lead to a more efficient response to emergencies [15,16].

Institutional trust exhibits both relational and instrumental dimensions, directly tied to the outcomes of interactions between citizens and institutions. It can be affected by factors such as individual knowledge and the perceived ability of emergency workers [17]. Accordingly, institutional trust with respect to disaster management can be bolstered through the enhancement of public knowledge as a tool within emergency preparedness plans, and the perception of emergency workers’ competence (entailing confidence in their ability to devise effective strategies, engage with the community, and fortify community resilience) plays a pivotal role in fostering institutional trust [4]. Conversely, when the processes of forecasting and mitigating damages fail to be efficiently executed, there exists the potential for the erosion of trust among disaster victims [18]. A decline in institutional trust signifies the belief that when adverse events occur, institutions may not be relied upon to provide essential resources or to take actions aimed at ensuring safety and justice. Notably, this perception may endure over time and contribute to the perpetuation of psychological distress within the community [19].

DRR encompasses a trio of essential measures designed to address the challenges posed by disasters, namely coping, adaptation, and mitigation [20]. More precisely, (a) coping denotes the capacity of individuals, organizations, and institutions to effectively manage risk or disaster conditions through the utilization of their available skills and resources; (b) adaptation entails the process of making adjustments to current or anticipated risks and their corresponding impacts, with the aim of either mitigating harm or capitalizing on beneficial opportunities; (c) mitigation encompasses the procedure of reducing or minimizing the adverse consequences stemming from hazardous events [21]. 

Scholars concur that communities exhibiting higher levels of resilience demonstrate enhanced capacities for responding effectively to adversities, rendering them more adept at both preventing and managing disasters [22]. Resilience in the aftermath of a disaster is defined as the ability of a system, community, or society to resist, absorb, accommodate, adapt to, transform, and recover from the effects of a hazard in a timely and efficient manner, including the preservation and restoration of its essential basic structures and functions. It has been seen that resilience is closely related to levels of trust [23]. Also, the relationship between trust and environmentally responsible behavior has been extensively explored in the academic literature. Trust plays a crucial role in alleviating the cognitive challenges associated with evaluating risks and making corresponding behavioral judgments throughout all phases of DRR, as well as improving the quality and rapidity of decision-making [8]. Therefore, it might be of utmost importance to critically examine the role of trust in DRR.

### 1.1. Trust and Disaster Risk Reduction

Natural disasters exert substantial damage on a global scale, with communities often serving as the first responders, tackling the crisis before external assistance arrives [24]. Nevertheless, the literature on the subject underscores that disasters are not solely natural occurrences; rather, they are shaped by policy failures and inequities [25]. This makes DRR necessary for reducing the dramatic impact of disasters on communities and promoting resilience. Scholars concur that communities exhibiting greater resilience are more adept at responding effectively to various adversities [22]. It has also been observed that resilience is positively associated with the community’s levels of trust [23].

A comprehensive framework for DRR encompasses four key phases, namely prevention, preparedness, response, and recovery [26]. In the prevention phase, efforts are directed at safeguarding individuals from potential natural disasters, thus affording long-term protection. The preparedness phase precedes the occurrence of a disaster, wherein experts in risk assessment strive to minimize potential harm. The response phase entails actions taken in response to a significant catastrophe or emergency, with a focus on saving lives, reducing economic losses, and alleviating suffering. Lastly, the recovery phase aims to restore normalcy in community life and mitigate the aftermath and long-term effects of disasters following their subsidence [27]. It is imperative to properly manage each of these phases to strike a balance between the enhancement of community resilience, risk reduction, and ensuring the effectiveness of response and recovery capabilities. In fact, these four phases are not linear; they overlap and complement one another. For example, recovery efforts may commence during the response phase, and mitigation strategies and future plans for prevention and preparedness could be explored during the recovery phase [26].

In communities characterized by strong trust, solidarity, and active participation, responses to disasters tend to be more effective. Residents in such communities are inclined to assist neighbors in need by sharing resources, providing shelter, offering financial support, aiding in disaster preparedness through early warning information, and providing emotional support [28]. Furthermore, trust among community members can facilitate resource access planning and the establishment of rescue teams during the response phase [15], and a high level of trust in institutions and the government can expedite recovery following a disaster [16].

After a disaster, the destruction of dwellings and infrastructure and the loss of lives can result in the depletion of physical and human capital. Nonetheless, community collaboration in addressing challenges and providing aid measures has the potential to strengthen trust within the community and towards institutions [29]. When each phase of DRR is adequately managed, trust remains the least impacted factor and can serve as a source of resilience for the community [30] because it is cultivated over time through interpersonal exchanges and collective actions that may protect individuals from the traumatizing effects of the disaster event [31]. Consequently, interest in the concept of trust, its function, and its significance in disaster management continues to burgeon across various disciplines.

Prior studies have emphasized the role of trust in the different phases of DRR, encompassing prevention [32], preparedness [33], response [34], and recovery [35]. These studies shed light on the specific role of each phase on community reactions to disasters. However, these studies have predominantly focused on a limited number of communities during specific disasters, with few of them extending their examination of trust across entire cities or regions and scrutinizing the impact of trust across all disaster phases. Also, studies have primarily concentrated on specific types of trust (i.e., community or institutional). Furthermore, research methods for investigating trust have been diverse and sometimes inconsistent, often privileging quantitative or qualitative approaches without a clear rationale for the preferred research method. In fact, given the inherent challenge of assessing trust in DRR, it frequently emerges as an avenue for future research within the scientific literature [3].

The current critical review focuses on the importance of trust and its role throughout all phases of disasters. It emphasizes trust as a crucial variable in DRR, investigating how trust shapes community prevention, preparedness, response, and recovery in the face of natural disasters. Previous research has sought to understand the role of trust in each distinct phase of DRR. Furthermore, various studies provide diverse definitions and interpretations of trust in DRR and various well-being outcomes. The separate examination of trust in each phase yields a complex array of outcomes, indicating the necessity to summarize and synthesize this literature. Therefore, in this critical review of the literature on the significance of trust across all DRR phases, our objectives are to summarize study findings, systematize relevant knowledge, identify potential gaps in the existing literature, and outline future research directions in this domain.

### 1.2. Aims of the Study

We conducted an extensive examination of the extant literature pertaining to community and institutional trust in DRR to synthesize the study findings and systematize the relevant knowledge while also identifying potential gaps in the existing literature and delineating directions for future research in this field.

More specifically, the literature review was undertaken with the following objectives:(1)To elucidate the role of trust in the different phases of DRR, encompassing prevention, preparedness, response, and recovery;(2)To gain insight into the relationship between trust and community resilience in the management of disaster risk, with the aim of enhancing community responses and advancing strategies for bolstering community resilience.

## 2. Methods

We present here a rapid review of the subject of trust in the context of DRR. This review does not describe research findings in all detail [36], but it was undertaken with the primary objective of summarizing and structuring the existing knowledge related to community and institutional trust in DRR.

In order to achieve this objective, we conducted an analysis of the pertinent literature concerning trust in DRR. Initially, we scrutinized recent research on the topic to find studies using rigorous methods. Then, we selected the appropriate and relevant search keywords (e.g., trust, disaster risk reduction, resilience, prevention, preparedness, response, recovery, and community). This approach aimed to enhance and update the knowledge base in this field, ensuring its relevance to current and future cohorts. The keywords were chosen to encompass various facets of DRR and trust at the community and institutional levels. Subsequently, we performed searches in academic databases (PubMed/Medline, ISI Web of Science, and SCOPUS). Empirical studies that investigated the role of trust dynamics in DRR were deemed eligible based on the following inclusion criteria: (1) they employed a cohort, case–control, cross-sectional study, and/or experimental design. Publications were excluded if (1) they were not original articles (e.g., proceeding, review, opinion paper, or dissertation) and (2) they did not specifically focus on natural disasters. Thereafter, we scrutinized the retrieved literature to select articles of relevance, and further, we explored their reference lists to identify additional pertinent articles that had not been initially captured in the search but held significance for inclusion in the present review. All the selected articles were subjected to close scrutiny and were subsequently summarized to construct the foundation for this review.

## 3. Results

In the subsequent sections, we discuss the findings derived from the analysis of articles pertaining to trust in DRR, organized in accordance with the four DRR phases of prevention, preparedness, response, and recovery, as outlined in our review framework.

### 3.1. Selected Articles

The review encompasses a total of 40 studies that were documented in peer-reviewed journal articles, all of which delve into the subject of trust in DRR. Among these 40 studies, 23 adopted a quantitative research design, 10 employed qualitative research methods, and 7 were mixed-method studies or adopted alternative research designs (e.g., they were based on a participatory approach). These studies featured both male and female participants, were in English language, and were all published in peer-reviewed journals. Figure 1 summarizes the research methods of the selected studies. The geographical representation of these studies is diverse and spans a diverse array of regions, including Australia, Bangladesh, Brazil, Chile, China, Ecuador, England, France, India, Indonesia, Israel, Italy, Japan, Korea, Malta, New Zealand, Norway, Pakistan, Romania, Spain, Sweden, Taiwan, and the USA (e.g., California, Florida, Hawaii, Louisiana, and Texas). Therefore, trust dynamics discussed in these articles encompass research conducted in a variety of countries across South Asia, Oceania, North and South America, and Europe, listed on the basis of research conducted on the topic. Notably, there is a noticeable dearth of studies exploring trust dynamics within the MENA region and in Africa, Central Asia, and North Asia.

### 3.2. Trust in the Prevention Phase

The primary objective of the prevention phase in DRR is to safeguard individuals from potential natural disasters. Although the literature posits that disasters frequently stem from policy failures and inequities, rendering complete avoidance plausible, preventive strategies are designed to offer sustained protection, acknowledging the potential for their future occurrence [37]. This phase implies proactive approaches involving the identification of potential hazards and the formulation of precautionary measures to mitigate their impact. During this phase, the role of trust in risk management is paramount, with its efficacy depending on various factors, including leadership style and effective communication by authorities [38,39].

The studies exploring trust in the prevention phase predominantly utilized quantitative research design (*n* = 3), followed by qualitative research design (*n* = 2) and mixed methods studies or other research design (*n* = 2). 

Quantitative studies underscore a significant and positive relationship between trust in public authorities and the capacity of communities to effectively respond to disaster predictions. For example, individuals who exhibit high levels of trust in institutions before extreme natural events, such as hurricanes, earthquakes, or wildfires, tend to be better prepared to manage these events [20]. This underscores the importance of local confidence in the information provided by public authorities during the prevention phase, as it affords residents ample time for preparation, leading to both enhanced disaster management and reinforcement of institutional trust [20]. Additionally, Zhong and colleagues [40] highlighted the critical role of trust in natural hazard warnings and their sources in people’s understanding and application of risk information: if trust in hazard warnings and their sources is low, individuals are more likely to disregard early warnings, while they are more inclined to follow them when trust is present. Furthermore, Rana and colleagues [41] demonstrated that individuals within communities characterized by strong mutual trust exhibit a heightened motivation to undertake preventive actions prior to disasters; conversely, when trust in local government officials is lacking, the motivation for such actions diminishes.

Qualitative studies emphasize the significance of building relationships in communities as well as fostering community and institutional trust through diverse cross-sector collaborations and partnerships before disasters. These collaborations should involve various stakeholders, such as community members, faith-based organizations, academic institutions, hospitals, police, public health services, neighborhood associations, and government agencies. Such collaborations and partnerships in the prevention phase have indeed proven effective in tailoring disaster responses to the unique needs of specific communities and populations and addressing key challenges in DRR, including information gaps, service inadequacies, and resource limitations [34]. Torres and colleagues [42] observed that adaptive actions are more likely to occur when high levels of networking social capital, grounded in trust and reciprocity, exist within communities in conjunction with well-functioning agencies. This synergy enhances community learning in disaster risk management and underscores the role of trust in the optimal functioning and stability of the political system during a disaster. 

In another study utilizing a mixed research design, Odiase and colleagues [43] highlighted that communities lacking prior disaster experience and with limited previous exposure to natural hazards tend to exhibit increased trust in official and expert information. However, the impact of official information on risk perception, as well as the subsequent trust in this information, varies among different types of hazards and is intricately linked to how participants perceive the risks that could impact them. This underscores the critical importance of fostering community trust during the prevention phase, particularly when a natural disaster has not yet occurred.

Another study conducted by Pratama and Nurmandi [38] placed the focus on the importance of leadership in disaster prevention. The authors observed that the leadership style of institutions is significantly linked to the motivation for agreement about preventative actions in the community, thereby influencing community trust in the initial management of disasters. In general, the interdependencies among institutions, agencies, and organizations, facilitated through interactive processes such as face-to-face dialogues among their representatives, contribute to increasing community trust, fostering social capital, and cultivating a collaborative culture. These aspects substantially accelerate decision making and lead to fruitful collaborations in disaster risk management. Notably, the research findings highlighted that trust, particularly in terms of dependability and competence, is significantly associated with collaborative planning. Collaborative leadership enhances the reliability and expertise of the community network, and trust enriches the planning process and its outcomes. These findings suggest that the prevention processes should involve the leaders of the institutions engaged in disaster operations to ensure a successful outcome. 

In conclusion, scholarly investigations suggest that trust plays a crucial role in the prevention phase, offering enduring safeguards against disasters. Specifically, empirical studies demonstrate that individuals with elevated levels of trust in institutions prior to severe natural events are more inclined to heed early warnings and exhibit enhanced readiness to manage such occurrences. Also, trust plays a role in augmenting community awareness, cultivating a deeper comprehension of hazard processes, and facilitating interaction between individuals and organizations. Table 1 provides a summary of the key findings regarding the role of trust in the prevention phase of DRR.

### 3.3. Trust in the Preparedness Phase

The preparedness phase of DRR occurs prior to the onset of a disaster. During this phase, local law enforcement agencies, community leaders and authorities, and risk experts engage in activities aimed at understanding the potential impact of a disaster on the overall community and devising strategies to mitigate risk. This phase is characterized by a range of activities essential for community preparedness for natural disasters, including risk-related planning, organization, training, equipment provision, exercises, evaluation, and identification of corrective measures. The primary focus during preparedness is on understanding the potential consequences of disasters on the community and how information, education, and training can enhance the community’s capacity to respond to and recover from such events. Also in the preparedness phase, trust needs to be cultivated and plays a critical role for DRR.

The majority of studies on trust in the preparedness phase employed a quantitative research design (*n* = 7), followed by a qualitative research design (*n* = 4), and mixed-method studies or other research designs (*n* = 3).

Quantitative studies revealed that trust in public health departments and government is significantly associated with active engagement in resilience-building activities, with a particular focus on disaster preparedness [33,44]. Bodas and colleagues [45] highlighted that trust in government, local authorities, and the media acts as a predictor of individual preparedness and risk awareness. On the other hand, Paton and colleagues [46] suggested that a lack of institutional trust within communities can be attributed to authorities’ failure to provide the necessary information and resources that community members perceive as vital for their needs and expectations during the preparedness phase, which might account for the negative association they found between community participation and trust during this phase.

In terms of community trust, Goidel and colleagues [47] showed that social trust (the individual’s perception concerning trust within a community) is positively associated with perceptions of community preparedness and disaster risk awareness. Communities with higher levels of social trust tend to be more aware and engaged in planning and preparation activities before disasters, including precautionary actions such as disaster planning and evacuation. Additionally, Ma and colleagues [1] found a positive correlation between community trust and community participation behavior, emphasizing that community trust fosters interaction between individuals and organizations while enhancing motivations for community involvement in disaster prevention and mitigation. These authors also found a significant association between community trust and place attachment, that is, the positive emotional bond between people and their place of residence. Accordingly, building a positive emotional connection with a place instills feelings of safety and security among individuals, encourages long-term residence, and fosters a strong sense of trust.

Furthermore, Adams and colleagues [48] found a positive association between community trust and the successful planning of partnerships and capacity building within community and faith-based organizations. Specifically, capacity building encompasses a set of activities where local health departments collaborate with and provide training for community and faith-based organizations, enabling them to be prepared to assist their members, clients, or other citizens in times of disaster. Partnership planning involves community-based collaborative planning between local health departments and community- and faith-based organizations, focusing on the development and implementation of community-wide preparedness and response plans, participation in community drills, and ensuring ongoing coordination before and after the disaster event. This coordination proves to be indispensable in nurturing a sense of community trust.

Qualitative studies highlight that, in the preparedness phase, there may be increased awareness of risks, which can lead to a reduction in trust as individuals become aware of the complexity of emergency and disaster management [4]. Accordingly, Marin and colleagues [49] stressed the importance of disaster experts establishing collaborative and participatory relationships with local communities and decision makers. This serves not only to reduce the negative impact of risk awareness and uncertainty but also to increase community trust. They also emphasize the significance of integrating local and traditional knowledge as valuable information sources for disaster experts and scientists. Engaging with the local community and taking into account people’s knowledge can help foster community trust among stakeholders involved in DRR efforts. In this context, Thouret and colleagues [50] added that trust in government agencies is founded on personal experiences during past disasters and respect for authorities. Among the factors that instill trust and enable people to cope with disasters, they identified the relevant role played by associations and solidarity networks, as well as shared trust in local officials.

Also, Pendergrast and colleagues [51] conducted a qualitative study centered on elderly individuals. They suggested that aging-in-place organizations can play a crucial role in promoting institutional trust; in fact, these organizations may serve as trusted sources of disaster-related information and provide valuable insights into the appropriateness of disaster plans for older adults. They also recommended some strategies for implementing disaster preparedness for this specific population, and proposed peer-to-peer educational programs in which old adult members or volunteers offer formal or informal education on disaster preparedness and response as especially effective interventions. Accordingly, they emphasize the importance of bidirectional relationships, cultural awareness, and trust in disaster risk communication for effective preparedness for risk. 

The capacity to develop effective collaborations and partnerships at different levels—among the individuals within a community, between community members and institutions, and among different institutions—is thus critical for the preparedness phase of DRR. In a mixed-methods study, Kitagawa [52] found that developing and delivering preparedness programs based on collaborative partnerships among a diverse range of stakeholders leads to the establishment of empowering and trustworthy relationships between community members and authorities; through their collaboration, both parties become acquainted with each other, exchange ideas and information, and cooperate in developing preparedness plans. Moreover, Muller and colleagues [53] showed that low levels of trust and negative relationships with other communities can hinder the development of effective disaster networks within a community. Such networks are crucial for providing critical information and resources before, during, and after a disaster. Thus, in the preparedness phase, authorities should also establish positive connections between different communities to foster community trust and effectively manage the risk. 

In this context, Stone and colleagues [39], adopting a community-based approach involving the active participation of community members, emphasized the significance of establishing trust-based relationships among citizens, scientists, and civil protection authorities. Active participation indeed plays a pivotal role not only in the preparedness phase but also in bolstering the overall resilience of the community. Through effective utilization of communication channels, trust contributes to enhancing community awareness and fostering a deeper understanding of hazard processes, and thus, it plays a fundamental role in increasing community preparedness. Furthermore, trust can serve as the cornerstone for developing effective early warning systems for residents. In fact, in uncertain circumstances, high levels of trust between residents and scientists facilitate the direct dissemination of critical information for risk management; specifically, this collaborative relationship between scientists and residents incentivizes community members to undertake risk-reducing actions that are informed via scientific information: for example, when individuals receive evacuation recommendations from a trusted source, whether through informal, direct communication channels or official mediums such as national radio or television, they are more inclined to make swift and appropriate decisions [39]. 

In summary, research indicates that trust emerges as a significant factor during the preparedness phase, facilitating community involvement in activities such as response planning and evacuation. Additionally, trust assumes a crucial role in cultivating increased motivation for community engagement in disaster preparedness and mitigation initiatives at both the individual and community levels. Table 2 provides a summary of the key findings regarding the role of trust in the preparedness phase of DRR.

### 3.4. Trust in the Response Phase

The response phase describes the reaction to a significant catastrophe or emergency, involving measures aimed at saving lives, minimizing economic losses, and alleviating suffering. This phase encompasses the coordination and management of human and physical resources, with a primary focus on ensuring the safety of the community. Given that traumatic events evoke high-intensity negative emotions [54], which are predictive of psychopathological outcomes [55], it becomes crucial to instill a positive feeling of trust among individuals during this difficult phase. Trust can indeed serve as a fundamental factor in mitigating the psychopathological effects associated with disasters.

The majority of studies examining trust in the response phase adopted a quantitative research design (*n* = 6), followed by a qualitative research design (*n* = 2), and one mixed method study.

Quantitative research primarily emphasizes the significance of social capital, where trust plays a pivotal role in effectively managing disasters during the response phase. Specifically, social capital indicates the networks of relationships among people who live in a particular society that enable that society to function effectively. Social capital is thus represented by community trust and reciprocity, and it implies shared values, a sense of community, and a network of communications [56]. 

In this respect, Carone and colleagues [57] observed that being exposed to a disaster might even foster trust and social cohesion, encouraging enhanced collaboration among citizens and between citizens and institutions. This is contingent upon the effectiveness of emergency communication and citizens’ trust in the communicator during disasters. Given the positive influence of reliable communication on community resilience, an improvement in emergency communication during the response phase can have a beneficial impact on both community trust and overall social resilience. For example, Slack and colleagues [58] identified that a lack of trust in institutions can create a disconnection between expert and community risk assessments. Nevertheless, confidence and trust in institutions’ ability to manage risk are essential in modern society, where individuals often lack comprehensive knowledge of the threats posed by various hazards. Accordingly, Choo and Yoon [59] highlighted the role of social capital, including trust, in determining the response capacity of a community. Communities characterized by strong trust among their residents and in public institutions exhibit higher citizen participation and a robust disaster response capacity, ultimately leading to quicker recovery from disasters. Furthermore, Dvir and colleagues [60] showed that individuals who have greater trust in the government’s ability to provide necessary services when needed tend to be less anxious during emergency situations. These researchers also explored demographic factors influencing institutional trust, revealing that age is associated with institutional trust, with older citizens expressing greater concern and lower trust in their expected behavior during risky situations; gender also seems to play a role in the response phase, as women report higher levels of concern than males, particularly in scenarios requiring actions like evacuation. 

Additionally, Aldrich [61] underscored the role of social infrastructure as the foundation for civic engagement and community trust, particularly in disaster response and recovery. Spaces and places that facilitate community interactions and activities, such as community centers, libraries, walking trails, and faith-based spaces, serve as the building blocks of social capital, including community trust. In a similar vein, Faisal and colleagues [62] reported that residents with a positive perception of the institutional response to risk and access to institutional services and infrastructure are more likely to engage in a greater number of adaptation strategies for economic and ecological benefits, also contributing to the development of a stronger sense of community trust.

Qualitative studies predominantly concentrate on institutional trust. For instance, Lo and colleagues [63] showed that higher-level authorities, such as the national government, garner increased trust when they convincingly demonstrate their competence in effectively managing major natural hazards, in contrast to residential committees or more general community entities. Conversely, Moreno and colleagues [64] accentuate the central role of community resilience in the response phase, particularly when external aid is scarce. They highlight how communities are proactive agents, with their levels of trust and resilience capacities playing a pivotal role in ensuring the community’s survival immediately following a disaster. Enhancing a community’s resilience in the face of critical events necessitates a high degree of trust, signifying that competent members of society are poised to provide assistance. The foundations of trust and trustworthiness rest upon the competence of institutional members in responding to events, maintaining transparent communication, demonstrating genuine concern for citizens, and ensuring that the quality of services aligns with community expectations.

A unique study using a mixed research design delved into the concept of distrust in authorities. Specifically, Appleby-Arnold and colleagues [65] suggested that distrust in authorities might originate from negative personal experiences and unmet expectations during the response phase, potentially leading to a negative climate in the relationship between citizens and authorities during the response phase and potentially establishing a cycle of distrust for future risks. 

In summary, research indicates that during the response phase, communities displaying greater levels of community and institutional trust exhibit a more robust capacity for disaster response. Also, the efficacy of emergency communication is linked to levels of trust. Individuals who manifest heightened trust in the government’s capacity to deliver essential assistance when needed tend to encounter diminished levels of apprehension and demonstrate enhanced responsiveness in moments of emergency. Table 3 summarizes the main findings regarding the role of trust in the response phase of DRR.

### 3.5. Trust in the Recovery Phase

The recovery phase initiates promptly following the cessation of a disaster, aiming to restore a sense of normality to the affected areas and individuals while implementing mitigation strategies to reduce the potential impact of future catastrophes and emergencies. Within this phase, communities dedicate their efforts to the restoration of resources, infrastructure, and activities that endured the effects of the disaster, with trust assuming a pivotal role in facilitating this restorative process.

Specifically, trust can serve as a psychological factor that helps in post-disaster reconstruction and recovery, particularly if the preceding phases of DRR have been effectively managed. In fact, trust evolves over time as a result of recurring positive social interactions within communities and through ongoing positive actions by institutions that are remembered even in the wake of a traumatic event [31]. 

Consequently, multiple studies have delved into the significance of trust during the recovery phase. Most of the studies examining trust in the recovery phase employed a quantitative research design (*n* = 7), followed by a qualitative research design (*n* = 2), and one mixed method study. 

A quantitative study conducted by Bhandari and colleagues [66] showed that disaster resilience in the recovery phase is closely linked to hazard awareness—a factor that is intrinsically connected to trust since hazard awareness is linked to adequate communication from trusted institutions in previous DRR phases. This finding supports the notion that heightened cooperation during the recovery phase contributes to greater trust among community members and institutions. Conversely, Gero and colleagues [31] identified that after an earthquake, the arrival of internally displaced persons in another community led to a decline in both generalized and local trust. Their study showed that forced migration following a disaster, even within the same city from one district to another, could erode trust among non-relocated residents, both in relation to individuals from other communities and those from the same community. Building trust among community residents hinges on repeated social interactions over an extended period, while exposure to outsiders or outgroups may trigger conflicts and mistrust [67]. Thus, it seems critical to prepare communities before relocation in previous DRR phases, fostering their reciprocal knowledge, interactions, and exchanges.

Furthermore, Joerin and colleagues [35] underscored the importance of the interplay between the community and authorities in the recovery process. Ongoing vulnerabilities, such as weak cohesion between communities and institutions, may hinder the establishment of active and trusting relationships between residents and local authorities. Thus, the ability of all stakeholders, including local authorities, the community, the private sector, academia, and non-governmental organizations, to lead, trust, and communicate with each other is considered essential during the recovery process. This perspective is also supported by Zander and colleagues [68], who argued that when the government provides emergency information that meets the public’s needs during the recovery phase, the citizens tend to place more trust in authorities and are less likely to seek information from other sources. 

Antronico and colleagues [69] further suggested that those who trust the ability of local policymakers to handle disasters are also likely to trust the national and international political authorities in the fight against disasters. Conversely, those who do not trust local authorities may not trust the capacity of national and international political authorities to address the complexities of disaster management and DRR. Notably, such an absence of trust in political institutions may represent a perceived barrier associated with the lack of action to mitigate climate change. 

Other studies have focused on psychological conditions related to trust following a disaster. Matthews and colleagues [70] found that higher levels of perceived social cohesion, encompassing individual subjective perceptions of belonging, community trust, generalized reciprocity, and optimism, were significantly associated with lower levels of post-traumatic stress disorder symptoms following a disaster. Accordingly, community trust, a sense of belonging to the community, and optimism were significantly associated with reduced distress in people during the recovery phase. Additionally, Thoresen and colleagues [2] found that chronic negative consequences following a disaster can result from poor communication between non-responsive authorities and disappointed victims. Individuals who have personal experiences with untrustworthy authorities in situations of strong negative emotional impact may never regain their trust in these authorities. This loss of trust can represent a significant resource loss, also reducing people’s ability to recover.

In a qualitative study, Ching and colleagues [71] emphasized that a crucial aspect of resilience after a disaster is strengthening relationships among neighbors, non-government organizations, and government agencies to establish a strong foundation of trust and cohesion within the community and among its partners. Regarding the elements that can enhance trust at this stage, Parkinson and colleagues [72] highlighted the importance of providing appropriate emotional support during the recovery process to build community trust. Such support encompasses medical interventions (e.g., prescriptions for antidepressants when necessary) and psychological interventions (e.g., psychological therapies) and extends to workplace counseling, professional guidance, group support, community-based assistance, and self-help.

Lastly, in a mixed-method research design, He and colleagues [73] emphasized that householders’ recovery experiences influenced patterns of trust within local communities, also shaping people’s expectations regarding future disaster management. Positive experiences during the recovery process fostered community trust and confidence in others, whereas negative experiences tended to erode trust and generate uncertainty. 

In summary, research findings indicate that during the recovery phase, trust assumes a pivotal role in strengthening connections among citizens, non-governmental organizations, and government agencies. The existence of trustful relationships among these entities is essential for bolstering individuals’ capacity to recover and improving their subsequent resilience to risks. Table 4 provides a summary of the main findings concerning the role of trust in the recovery phase of DRR.

## 4. Discussion

This review has examined research focusing on trust within the context of DRR to address prevalent research gaps. These gaps include the need to (1) comprehend the role of trust across the various phases of DRR, namely prevention, preparedness, response, and recovery, and (2) elucidate the interconnection between trust and community resilience during natural disasters to enhance community responses and promote more effective strategies to foster community resilience.

Although differences in the definition and measurement of trust create challenges in drawing uniform conclusions from the body of research, the studies consistently identify a critical role of trust in all aspects of DRR. The findings from this critical review indicate that community and institutional trust serve as vital resources that aid communities in the management of natural disasters and contribute to the enhancement of community resilience.

Concerning the prevention phase, it is evident that a high level of trust in information provided by public authorities during this stage is critical, as it allows residents to prepare for disasters [20]. Conversely, a lack of trust in authorities responsible for disseminating early warnings and emergency information represents a significant hindrance to community resilience [74]. 

In studies concerning the preparedness phase, the close connection between trust and DRR becomes even more evident. More precisely, when institutions are perceived as reliable and community members have increased access to information and resources for preparedness, this empowers community members to better prepare during this phase [4,48]. As a result, the level of trust in institutions significantly impacts the degree to which the citizens actively participate in disaster prevention and preparedness measures [33].

In the response phase, the importance of social capital—of which trust is a pivotal component—and social infrastructures are emphasized. Social infrastructures, which provide spaces for community members to convene during disasters, constitute the underpinnings of civic engagement and community trust [61]. 

Regarding the recovery phase, it is evident that heightened collaboration among citizens, stakeholders, and institutions engenders trust in the aftermath of a disaster [66]. Therefore, the greater the degree of cooperation during the recovery phase, the more trust is fostered among individuals and towards the institution. It becomes imperative for institutions to facilitate collaborative initiatives involving community members, faith-based organizations, academic institutions, hospitals, police, public health services, neighborhood associations, and government agencies [34]. 

Comparing the critical facets of trust delineated in the four phases of DRR, the initial two phases mainly exhibit trust levels influenced by the nature of the received information and the credibility of its source. Conversely, in the latter two phases, trust levels are shaped by tangible aspects of the community, such as social infrastructure, combined with social elements like the degree of cooperation among actors (i.e., citizens, associations, non-governmental agencies, and institutions). It appears that the evolution across the four phases of DRR entails a shift from more abstract to more concrete aspects, all the while underscoring the significance of both in the entire risk management process. However, it is important to point out that the levels of trust in the first phases may reflect the levels of trust in the last phases, indicating the need to establish a robust foundation of trust to enhance the community’s overall resilience in the face of risks.

Developing a more comprehensive framework to elucidate the mechanisms of trust in the different phases of DRR requires a thorough understanding of research findings. In fact, when the results of the review are comprehensively examined, a recurring theme emerges: throughout the entire disaster management cycle, individuals can cultivate trust in their community (e.g., relatives, neighbors, coworkers) and institutions (e.g., agencies, authorities, government). Establishing trust can significantly increase the capacity for managing disaster risks and enhancing resilience within communities. By relying on trust, individuals, communities, and institutions have the capacity to absorb and recover from disasters, concurrently fostering positive adaptation and transformation in their behaviors amid enduring changes and uncertainties. Consequently, there is a crucial need to promote collaborations and partnerships across various sectors, prioritizing community engagement and endorsing leadership models capable of expanding trust in the disaster management process. This, in turn, initiates a reinforcing cycle leading to an elevated level of trust in institutions. Moreover, the importance of sufficient information and communication is emphasized throughout the entire cycle. Research indicates that when authorities provide emergency information aligned with citizens’ needs, this bolsters trust in authorities, thereby reducing the likelihood of seeking information from alternative and unreliable sources. Conversely, inadequate communication by authorities is associated with enduring negative consequences in disaster management.

It is worth noting that institutional efforts to foster trust and promote community resilience should be initiated even before the initial stages of risk management. Since the four phases of DRR are interconnected and often overlap [26], effective DRR measures demand an early implementation of trust. This is because establishing a robust sense of trust is paramount for limiting damage and expediting recovery, as shown by research indicating that communities characterized by strong trust, solidarity, and active participation exhibit more efficient responses to natural disasters.

Specific actions can be taken to cultivate trust at various phases of DRR, thus ensuring effective coping with natural disasters. In the prevention phase, it might be crucial to disseminate updated information and training materials on natural disaster management to community members, as well as through the most widely used institutional and social channels. Actions undertaken in this phase might align with those in the preparedness phase, wherein organizing information-sharing sessions, like public meetings or focus groups on natural disaster management topics, is recommended. Inviting community leaders and local law enforcement officers to address community gatherings on diverse themes, including the community’s role in natural disaster management, might also be crucial. Equally important is involving the community in implementing natural disaster management plans during the response phase, ensuring community engagement in appropriate ways when executing these plans. In this phase, it is imperative to utilize popular communication channels (e.g., phones, social media, and television) to communicate promptly with the community regarding DRR actions. For instance, social media platforms may serve as prime sources of real-time information during emergencies [68]. During the recovery phase, regular communication regarding the progress and performance of recovery plans should be maintained, giving due credit to the community for its contributions and efforts in mitigating the effects of disasters. Consistent and clear communication is indispensable for keeping citizens informed and fostering trust within the community. A lack of transparency, even if unintentional, sows seeds of mistrust and misinformation. Through proficient use of communication channels (e.g., mass media or social media), experts in disaster management can facilitate the dissemination of emergency information and the cultivation of trust. 

Consequently, trust contributes to augmenting community awareness and fostering a more profound comprehension of hazard processes, playing a pivotal role in bolstering community resilience [68]. In the realm of disaster management, institutional policies should strive to enhance individuals’ trust in their community and toward institutions. The establishment of trust holds the potential to significantly fortify the capacity for disaster risk prevention, management, and resilience in communities.

Therefore, the findings reported in this review indicates that the levels of trust are intrinsically linked to a community’s resilience. Specifically, studies have indicated that trust serves as a direct predictor of resilience [45,57]. Similar to trust, resilience to disasters extends across all phases of a disaster (pre-event, during a disaster, and post-event phases) and is not confined solely to the disaster response and recovery stages [75]. Resilience is thus contingent on the community’s capacity to recover post-disaster and is also connected to the extent of resources and capabilities the community possesses, both before and after a disaster. An integral dimension of resilience involves fortifying relationships among citizens, NGOs, and government agencies, which are indispensable for establishing a robust foundation of trust and cohesion within the community and its collaborative partners when faced with disasters. Consequently, resilience and trust appear to be reciprocally influenced and are interdependent during disasters.

## 5. Conclusions

In this article, we have critically examined the role of community and institutional trust throughout all phases of DRR to summarize research findings in this field and gain a deeper understanding of the connection between trust and community resilience in disaster risk management. Consequently, this critical review has represented an initial endeavor to consolidate and systematize the available evidence concerning the impact of trust on DRR. The findings indicate that trust plays a pivotal role in every phase of DRR, influencing the efficacy of prevention actions by governments and individual preparedness, response, and recovery from disasters. Therefore, it is imperative to cultivate a “culture of trust”, both at the community and institutional levels, to effectively address the risks associated with disasters.

## 6. Limitations and Future Directions

The review comes with some limitations. Firstly, it includes the evaluation of trust in phases of DRR as they are treated independently. This is necessitated by data constraints, including the lack of longitudinal studies, but it may obscure the intricate interconnections and interdependencies between the phases. Secondly, the review exclusively considers the scientific literature, potentially disregarding valuable insights present in the gray literature. Thirdly, the review summarizes studies that employ various methods for assessing trust and its correlates in DRR, including ad hoc and validated instruments, which limits the possibility of comparing the results of the studies. Ultimately, this review failed to account for variations in the levels and types of trust (both community and institutional) between developed and underdeveloped countries, a phenomenon that could be shaped by contextual factors. Its limitations notwithstanding, the review clearly underscores that both institutional and community trust are linked to effective DRR initiatives.

The absence of longitudinal evidence that can demonstrate causal relationships between trust dynamics and individuals’ attitudes and behaviors in DRR should not be misconstrued as evidence of non-existence. It is imperative to conduct further research to unearth the causal connections between trust and effective DRR initiatives, a task that will likely necessitate the implementation of innovative research methodologies, such as longitudinal and experimental designs, capable of delving into the specific characteristics that influence trust in the context of DRR. 

The ultimate objective of such research efforts is to establish and disseminate a “culture of trust” within the communities. Constructing a culture of trust poses challenges due to the complex nature of the concept and the associated requirements for its implementation. Nevertheless, as with many complex subjects, transparent and open communication stands as the most potent tool that every community can wield. Honesty in communication regarding the necessity and practice of DRR can alleviate citizens’ fear and reduce inappropriate responses when the time comes to implement emergency management strategies. This culture of trust is indispensable for fostering community resilience and creating a bond wherein all community members collectively work towards safeguarding everyone’s safety [32].

## Figures and Tables

**Figure 1 ijerph-21-00029-f001:**
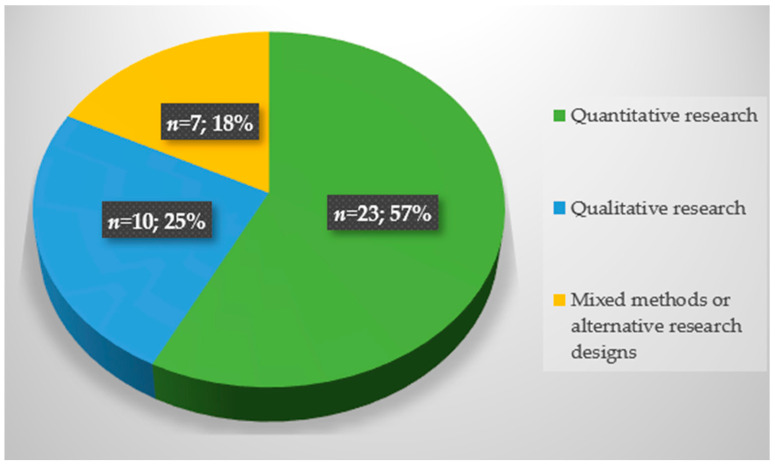
Distribution of included studies.

**Table 1 ijerph-21-00029-t001:** Themes related to trust in the prevention phase.

Authors	Type of Study	Country	Type of Trust (Institutional and/or Community)	Themes Identified for Trust in Prevention Phase
Alves et al., 2021 [20]	Quantitative	Brazil	Institutional	Trust in public authorities is correlated with communities’ capacity to cope with challenges.
Odiase et al., 2020 [43]	Mixed-method design	New Zealand	Institutional	Trust in official and expert information is influenced by prior experiences with disasters.
Pollock et al., 2019 [34]	Qualitative	United States	Community	Both community trust and institutional trust are integral to fostering inter-sector collaborations and partnerships in disaster prevention.
Pratama and Nurmandi, 2020 [38]	Mixed-method design	Indonesia	Community	Leadership style in the disaster management process has the potential to foster community trust.
Rana et al., 2020 [41]	Quantitative	Bangladesh	Community	Community trust is linked to the implementation of preventive measures prior to a disaster.
Torres et al., 2018 [42]	Qualitative	United States	Institutional	Institutional trust facilitates the effective functioning of government entities in the disaster prevention phase.
Zhong et al., 2021 [40]	Quantitative	China	Institutional	Trust in hazard warnings and their sources is associated with individuals’ comprehension and utilization of risk information.

**Table 2 ijerph-21-00029-t002:** Themes related to trust identified during the preparedness phase.

Authors	Type of Study	Country	Type of Trust (Institutional and/or Community)	Themes Identified for Trust in Preparedness Phase
Adams et al., 2017 [44]	Quantitative	United States	Institutional	Trust in public health departments is associated with higher levels of engagement in disaster preparedness activities.
Adams et al., 2018 [48]	Quantitative	United States	Community	Community trust is positively associated with collaborative planning and capacity building conducted by community and faith-based organizations.
Bian et al., 2022 [33]	Quantitative	China, Taiwan	Institutional	Trust in the government is linked to greater involvement in disaster preparedness.
Bodas et al., 2022 [45]	Quantitative	Italy, Romania, Spain, France, Sweden, Norway, Israel, Japan	Institutional	Trust in the government and local authorities serves as a predictor of individual preparedness.
Goidel et al., 2019 [47]	Quantitative	United States	Community	Social trust is closely intertwined with community preparedness and disaster awareness.
Humann et al., 2022 [4]	Qualitative	England	Community	An increase in awareness about the risks could potentially erode community trust.
Kitagawa, 2018 [52]	Mixed-method design	Japan	Institutional	Trust is fostered through programs that are developed based on collaborative partnerships involving a diverse range of disaster management stakeholders and the community.
Ma et al., 2022 [1]	Quantitative	China	Community	Community trust is positively correlated with engagement in community participation behaviors.
Marin et al., 2020 [49]	Qualitative	England	Institutional	Engaging with the local community, leveraging people’s knowledge, and utilizing their data-gathering capacity can help foster trust within the stakeholder community involved in DRR efforts.
Muller et al., 2014 [53]	Mixed-method design	United States	Community	Low levels of trust and limited relationships among communities may hinder the establishment of resilient disaster networks.
Paton et al., 2010 [46]	Quantitative	New Zealand, Japan	Institutional	The absence of institutional trust is connected to the authorities’ failure to provide the necessary information and resources during the preparedness phase.
Pendergrast et al., 2021 [51]	Qualitative	United States	Institutional	Building trust within the elderly community can be achieved by providing information on disaster preparedness.
Stone et al., 2014 [39]	Community-based approach	Ecuador	Institutional	Trust in disaster risk management authorities enhances community disaster awareness through effective communication channels.
Thouret et al., 2022 [50]	Qualitative	Indonesia	Institutional	Trust in government agencies is grounded in personal experiences during previous evacuations.

**Table 3 ijerph-21-00029-t003:** Themes related to trust in the response phase.

Authors	Type of Study	Country	Type of Trust (Institutional and/or Community)	Themes Identified on Trust in Response Phase
Aldrich, 2023 [61]	Quantitative	Japan	Community	Social infrastructure forms the basis upon which civic engagement and trust are built.
Appleby-Arnold et al., 2021 [65]	Mixed-method design	Romania, Malta	Institutional	Distrust in authorities can stem from personal experiences and unmet expectations during a response phase.
Carone et al., 2019 [57]	Quantitative	Italy	Institutional	Trust can be established through effective emergency communication during a disaster.
Choo and Yoon, 2022 [59]	Quantitative	Korea	Community Institutional	Communities with strong community and/or institutional trust show an adequate capacity to respond to disasters.
Dvir et al., 2022 [60]	Quantitative	United States	Institutional	Those who place greater trust in the government tend to be less anxious during times of emergency.
Faisal et al., 2021 [62]	Quantitative	Pakistan	Community	Residents with access to institutional services tend to develop higher levels of trust.
Lo et al., 2016 [63]	Qualitative	China	Institutional	Higher levels of trust are associated with the persuasive abilities of governements.
Moreno et al., 2019 [64]	Qualitative	Chile	Institutional	Trust hinges on institutional members’ responsiveness to an event and their perceived competence in managing the task.
Slack et al., 2020 [58]	Quantitative	United States	Institutional	The lack of trust in major institutional actors can create a gap between expert and community assessments of risk.

**Table 4 ijerph-21-00029-t004:** Themes related to trust in the recovery phase.

Authors	Type of Study	Country	Type of Trust (Institutional and/or Community)	Themes Identified for Trust in Recovery Phase
Antronico et al., 2020 [69]	Quantitative	Italy	Institutional	Trust in local policymakers’ ability to respond to an extreme event is intertwined with trust in the national and international political classes.
Bhandari et al., 2010 [66]	Quantitative	Japan	Community Institutional	The extent to which people collaborate during the recovery phase is associated with the level of trust among individuals and their trust in the institution.
Ching et al., 2020 [71]	Qualitative	United States	Community Institutional	Trust within the community and institutions is cultivated by enhancing relationships among neighbors, non-governmental organizations (NGOs), and government agencies.
Gero et al., 2020 [31]	Quantitative	Japan	Community	Building trust among community residents is linked to sustained social interactions over an extended period.
He et al., 2021 [73]	Mixed-method design	New Zealand	Community	Householders’ recovery experiences influence patterns of trust within local communities.
Joerin et al., 2018 [35]	Quantitative	India	Institutional	The ability of the institution to lead, trust, and communicate is considered fundamental during the recovery process.
Matthews et al., 2020 [70]	Quantitative	Australia	Community	Community trust, a sense of belonging, and optimism were significantly correlated with lower levels of distress.
Parkinson et al., 2022 [72]	Qualitative	Australia	Community	Offering appropriate emotional support during the recovery process is valuable for building community trust.
Thoresen et al., 2018 [2]	Quantitative	Norway	Institutional	The levels of institutional trust are lower among victims compared to the general population, with chronic negative consequences.
Zander et al., 2022 [68]	Quantitative	Australia	Institutional	Trust in authorities is connected to the emergency information provided by those authorities.

## Data Availability

Not applicable.

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
