# Peer review of "The Role of Trust in Disaster Risk Reduction: A Critical Review"

_ijerph, 2023, doi:10.3390/ijerph21010029_

Round 1

Reviewer 1 Report

Comments and Suggestions for Authors

Review of “The role of trust in disaster risk reduction:

A critical review”

(Manuscript Number: ijerph-2715292)

General comments: Generally, this study attempted to answer the following two important questions: 1) dedicate to the examination of research concerning community and institutional trust in the field of disaster risk reduction; 2) elucidate the influence of trust throughout the distinct phases of natural disaster management, namely prevention, preparedness, response, and recovery. This study is meaningful both theoretically and practically for the development of disaster management and disaster risk reduction. Nevertheless, several concerns should be addressed properly before the paper can be published in the International Journal of Environmental Research and Public Health.

First, the rationale for conducting this study was unclear, please make efforts to revise the introduction section, try to set the problem discussed in this paper in more clearly. For instance, before the “Aims of the study”, the reasons for conducting this study are not yet sufficient. And in the literature, please provide more evidence to address the purpose of this study is critical. In addition, before the “Trust and disaster risk reduction” section, a large section discussed the factors that affect trust, such as residential stability, ethical homogeneity, etc. This is confusing, as this information does not seem to be the focus of this study. Moreover, in the last paragraph before the “Trust and disaster risk reduction” section, this study proposed institutional trust. This looks very abrupt. Furthermore, the reason for exploring the relationship between trust and community resilience in the introduction section is not yet clear.

Second, in the methods section, the relevant content needs to be supplemented and clarified. For instance, why choose “resilience” as a search keyword? Reasons need to be explained for selecting search keywords. Moreover, what criteria are used for selecting “the selected articles of relevance”? Please elaborate on the screening process. (Line 176-180, Page 4 “Subsequently, we scrutinized the retrieved literature to select articles of relevance, and further, we explored their reference lists to identify additional pertinent articles that have not been initially captured in the search, but held significance for inclusion in the present review. All the selected articles were subjected to close scrutiny and were subsequently summarized to construct the foundation for this review.”). Furthermore, in the “Selected articles” section (Line 185-200, Page 4-5), it is recommended to use graphs and tables to display the research results.

Third, in the “results” section, it is recommended to add research summaries to each section. The “results” section elucidated the role of trust in the different phases of DRR, encompassing prevention, preparedness, response, and recovery. It is recommended to add research summaries to each part. Not just using tables. In addition, in Table 1-Table 4, the study categorizes trust types as institutional or community without any basis or explanation. Moreover, in Table 1-Table 4, the study divided the country of the selected articles. However, this point was not discussed in the discussion section. Are there any differences in the research conducted in different countries?

Fourth, in the discussion section, the relevant content needs to be supplemented and clarified. The study aims to elucidate the role of trust in the different phases of DRR, encompassing prevention, preparedness, response, and recovery. So, are there any similarities, differences, or connections between the roles of trust in the prevention, preparation, response, and recovery stages? In addition, another research purpose of this study is to gain insight into the relationship between trust and community resilience in the management of disaster risk. What is community resilience? This concept also needs further clarification. In the discussion section, the relationship between trust and community resilience needs more discussion. The current discussion is somewhat scattered.

Specific comments: 

1. Line 51-52, Page 2: “Consequently, trust transcends individual boundaries and maintains a close association with the concept of community.”

Please provide supporting references or evidence.

2. Line 64-67, Page 2: “Prior research suggests that community trust is influenced by two key factors, namely residential stability and ethnic homogeneity, which are construed as products of shared meanings and associated traditions that serve to define, inform, and shape how individuals make experience of the community and their collective sense of community[9,10]”

Actually, I do not see any necessity to mention this information in the Introduction section.

3. Line 151-152, Page 4: “In fact, given the inherent challenge of assessing trust in DRR, it frequently emerges as an avenue for future research within scientific literature.”

Please provide supporting references or evidence.

4. Line 173-176, Page 4: “Initially, we performed searches in academic databases (Pub-Med/Medline, ISI Web of Science, and SCOPUS), using a combination of relevant search keywords (e.g., trust, disaster risk reduction, resilience, prevention, preparedness, response, recovery, community)”

It should be “Initially, we performed searches in academic databases (Pub-Med/Medline, ISI Web of Science, and SCOPUS), using a combination of relevant search keywords (e.g., trust, disaster risk reduction, resilience, prevention, preparedness, response, recovery, community).”

5. Line 196-198, Page 4: “Therefore, trust dynamics discussed in these articles encompass research conducted in a variety of countries across South Asia, Oceania, North and South America, and Europe, with a gradually decreasing geographical diversity.”

Is this statement(“with a gradually decreasing geographical diversity”) accurate?

6. The value and significance of this study was not clear. 

Comments on the Quality of English Language

It is noted that the manuscript needs careful editing by someone with expertise in technical English editing paying particular attention to English grammar, spelling, and sentence structure so that the goals and results of the study are clear to the reader.

Author Response

Response to Reviewer 1 Comments

General comments: Generally, this study attempted to answer the following two important questions: 1) dedicate to the examination of research concerning community and institutional trust in the field of disaster risk reduction; 2) elucidate the influence of trust throughout the distinct phases of natural disaster management, namely prevention, preparedness, response, and recovery. This study is meaningful both theoretically and practically for the development of disaster management and disaster risk reduction. Nevertheless, several concerns should be addressed properly before the paper can be published in the International Journal of Environmental Research and Public Health.

Introduction

First, the rationale for conducting this study was unclear, please make efforts to revise the introduction section, try to set the problem discussed in this paper in more clearly. For instance, before the “Aims of the study”, the reasons for conducting this study are not yet sufficient. And in the literature, please provide more evidence to address the purpose of this study is critical. In addition, before the “Trust and disaster risk reduction” section, a large section discussed the factors that affect trust, such as residential stability, ethical homogeneity, etc. This is confusing, as this information does not seem to be the focus of this study. Moreover, in the last paragraph before the “Trust and disaster risk reduction” section, this study proposed institutional trust. This looks very abrupt. Furthermore, the reason for exploring the relationship between trust and community resilience in the introduction section is not yet clear.

Response: Thank you for the suggestion. In the revised version of the manuscript, we have better clarified the theoretical rationale behind the review (RL 32-115):

In recent years, there has been an increasing scholarly interest in the examination of trust within the domain of disaster risk reduction (DRR)1–4. Researchers hailing from diverse disciplines, including economics, law, psychology, and social sciences, have collectively recognized trust as a critical factor impacting the level of community re-sponse when confronted with natural disasters.

Trust can be broadly defined as “a psychological state comprising the intention to accept vulnerability based upon positive expectations of the intentions or behavior of another”5. The concept of trust assumes a fundamental role in comprehending the dy-namics of social interactions among individuals and groups. Trust necessitates the willing acceptance of vulnerability by the trusting party when engaging with a trusted entity—whether it be an individual, a group, or an institution—without the presence of immediate guarantees or assurances regarding the motivations and behaviors of the trusted party6,7.

Consequently, trust transcends individual boundaries and maintains a close asso-ciation with the concept of community4,8. As elucidated by Walker and colleagues9, trust is essential for forming and maintaining connections between individuals who may not interact otherwise. In this context, trust can be considered as a relational con-struct that defines the quality of interpersonal communication between individuals or groups. It enables individuals to establish relationships within their communities and with institutions, and facilitates the exchange of support and assistance10. Specifically, community trust can be characterized as a fundamental attribute of communities that implies the presence of confidence in participation and the belief that a community possesses the capacity to resolve disputes and engage in collectively accepted public endeavors11.

Notably, trust exhibits the capacity to mitigate the adverse impacts of psycholog-ical distress experienced by marginalized or economically disadvantaged groups12. Conversely, numerous findings indicate that the community's vulnerability and resili-ence levels are affected by insufficient levels of trust and unsupportive governance structures, hindering access to resources and inclusion in DRR processes. This, in turn, perpetuates a self-reinforcing cycle that erodes trust13.

Furthermore, it has been observed that trust in institutions, environmental groups, and scientists strongly correlates with the adoption of appropriate behaviors in DRR. These behaviors include individual-level preparatory actions, e.g., the acquisition of insurance and the endorsement of adaptation policies8,13. For this reason, in recent years, there has also been a growing interest in the examination of trust concerning government and institutional entities. The notion of institutional trust hinges upon the belief in the capability of institutions to effectively manage a diverse array of risks and social challenges14. This concept is frequently associated with the expectation that in-stitutions will implement policies that are advantageous and successful for the well-being of citizens, and is regarded as a significant gauge of broad-based political endorsement9.

Institutional trust assumes particular significance when risks and their potentially negative impact on citizens emanate primarily from sources beyond individual con-trol, as well as when the institution is entrusted with the responsibility of forecasting and addressing damages that are largely outside the control of citizens—such as in situations involving natural hazards, where citizens cannot predict the entity of dam-age and have little control over recovery processes. The significance of trust in reduc-ing complexities and costs of risks lies in fostering a robust sense of concern, solidarity, and active engagement within the community. This, in turn, may lead to a more effi-cient response to emergencies15,16.

Institutional trust exhibits both a relational and instrumental dimension, directly tied to the outcomes of interactions between citizens and institutions. It can be affected by factors such as individual knowledge and perceived ability of emergency workers17. Accordingly, institutional trust with respect to disaster management can be bolstered through the enhancement of public knowledge as a tool within emergency prepared-ness plans4, and the perception of emergency workers’ competence (entailing confi-dence in their ability to devise effective strategies, engage with the community, and fortify community resilience) plays a pivotal role in fostering institutional trust4. Con-versely, when the processes of forecasting and mitigating damages fail to be efficiently executed, there exists the potential for the erosion of trust among disaster victims18. A decline in institutional trust signifies the belief that when adverse events occur, insti-tutions may not be relied upon to provide essential resources or to take actions aimed at ensuring safety and justice. Notably, this perception may endure over time and con-tribute to the perpetuation of psychological distress within the community19.

DRR encompasses a trio of essential measures designed to address the challenges posed by disasters, namely coping, adaptation, and mitigation20. More precisely, (a) coping denotes the capacity of individuals, organizations, and institutions to effec-tively manage risk or disaster conditions through the utilization of their available skills and resources21; (b) adaptation entails the process of making adjustments to current or anticipated risks and their corresponding impacts, with the aim of either mitigating harm or capitalizing on beneficial opportunities21; (c) mitigation encompasses the pro-cedure of reducing or minimizing the adverse consequences stemming from hazardous events 21.

Scholars concur that communities exhibiting higher levels of resilience demon-strate enhanced capacities for responding effectively to adversities, rendering them more adept at both preventing and managing disasters22. Resilience in the aftermath of a disaster is defined as the ability of a system, community, or society to resist, absorb, accommodate, adapt to, transform, and recover from the effects of a hazard in a timely and efficient manner, including the preservation and restoration of its essential basic structures and functions. It has been seen that resilience is closely related to the levels of trust23. Also, the relationship between trust and environmentally responsible behav-ior has been extensively explored in the academic literature. Trust plays a crucial role in alleviating the cognitive challenges associated with evaluating risks and making corresponding behavioral judgments throughout all phases of DRR, also improving the quality and rapidity of decision-making8. Therefore, it might be of utmost importance to critically examine the role of trust in DRR.

[…]

This makes DRR necessary for reducing the dramatic impact of disasters on communi-ties and promoting resilience. Scholars concur that communities exhibiting greater re-silience are more adept at responding effectively to various adversities22. It has been also observed that resilience is closely associated with the community's levels of trust as well23.

[…]

The current critical review focuses on the importance of trust and its role throughout all phases of disasters. It emphasizes trust as a crucial variable in DRR, investigating how trust shapes community prevention, preparedness, response, and recovery in the face of natural disasters. Previous research has sought to understand the role of trust in each distinct phase of DRR. Furthermore, various studies provide diverse definitions and interpretations of trust in DRR and various well-being outcomes. The separate examination of trust in each phase yields a complex array of outcomes, indicating the necessity to summarize and synthesize this literature. Therefore, in this critical review of literature on the significance of trust across all disaster phases, our objectives are to summarize study findings, systematize relevant knowledge, identify potential gaps in existing literature, and outline future research directions in this domain.”

Also, the paper has been significantly restructured according to your comments, and definitions of each construct have been provided to increase clarity.

Methods

Second, in the methods section, the relevant content needs to be supplemented and clarified. For instance, why choose “resilience” as a search keyword? Reasons need to be explained for selecting search keywords. Moreover, what criteria are used for selecting “the selected articles of relevance”? Please elaborate on the screening process. (Line 176-180, Page 4 “Subsequently, we scrutinized the retrieved literature to select articles of relevance, and further, we explored their reference lists to identify additional pertinent articles that have not been initially captured in the search, but held significance for inclusion in the present review. All the selected articles were subjected to close scrutiny and were subsequently summarized to construct the foundation for this review.”). Furthermore, in the “Selected articles” section (Line 185-200, Page 4-5), it is recommended to use graphs and tables to display the research results.

Response: Thank you for the suggestion. In the revised version of the manuscript, we have better clarified the methods section behind the review (RL 196-207):

“Initially, we scrutinized recent research on the topic to find studies using rigorous methods. Then, we selected the appropriate and relevant search keywords (e.g., trust, disaster risk reduction, resilience, prevention, preparedness, response, recovery, community). This approach aimed to enhance and update the knowledge base in this field, ensuring its relevance to current and future cohorts. The keywords were chosen to encompass various facets of DRR and trust at the community and institutional level. Subsequently, we performed searches in academic databases (PubMed/Medline, ISI Web of Science, and SCOPUS). Empirical studies that investigated the role of trust dynamics in DRR were deemed eligible based on the following inclusion criteria: (1) they employed cohort, case-control, cross-sectional study, and/or experimental design. Publications were excluded if (1) they were not original articles (e.g., proceeding, review, opinion paper, or dissertation), and (2) they did not specifically focus on natural disaster.”

Results

Third, in the “results” section, it is recommended to add research summaries to each section. The “results” section elucidated the role of trust in the different phases of DRR, encompassing prevention, preparedness, response, and recovery. It is recommended to add research summaries to each part. Not just using tables. In addition, in Table 1-Table 4, the study categorizes trust types as institutional or community without any basis or explanation. Moreover, in Table 1-Table 4, the study divided the country of the selected articles. However, this point was not discussed in the discussion section. Are there any differences in the research conducted in different countries?

Response: Thank you for the suggestion. In the revised version of the manuscript, we have better clarified the results section behind the review:

“In conclusion, scholarly investigations suggest that trust plays a crucial role in the prevention phase, offering enduring safeguards against disasters. Specifically, empirical studies demonstrate that individuals with elevated levels of trust in institutions prior to severe natural events are more inclined to heed early warnings and exhibit enhanced readiness to manage such occurrences. Also, trust plays a role in augmenting community awareness, cultivating a deeper comprehension of hazard processes, and facilitating interaction between individuals and organizations.” (pg. 7; RL 305-311)

“In summary, research indicates that trust emerges as a significant factor during the preparedness phase, facilitating community involvement in activities such as response planning and evacuation. Additionally, trust assumes a crucial role in cultivating in-creased motivation for community engagement in disaster preparedness and mitiga-tion initiatives at both the individual and community levels.” (pg. 10; RL 424-428)

“In summary, research indicates that during the response phase, communities displaying greater levels of community and institutional trust exhibit a more robust capacity for disaster response. Also, the efficacy of emergency communication is linked to the levels of trust. Individuals who manifest heightened trust in the government's capacity to deliver essential assistance when needed tend to encounter diminished levels of apprehension and demonstrate enhanced responsivity in moments of emergency.” (pg. 12; RL 502-507)

“In summary, research findings indicate that during the recovery phase, trust assumes a pivotal role in strengthening connections among citizens, non-governmental organizations, and government agencies. The existence of trustful relationships among these entities is essential for bolstering individuals' capacity to recover and improving their subsequent resilience to risks.” (pg. 15; RL 589-593)

Regarding the differences of the trust in the research conducted in different countries, this review was unable to capture the differences between countries. This aspect has been integrated in the limitations section.

Discussion

Fourth, in the discussion section, the relevant content needs to be supplemented and clarified. The study aims to elucidate the role of trust in the different phases of DRR, encompassing prevention, preparedness, response, and recovery. So, are there any similarities, differences, or connections between the roles of trust in the prevention, preparation, response, and recovery stages? In addition, another research purpose of this study is to gain insight into the relationship between trust and community resilience in the management of disaster risk. What is community resilience? This concept also needs further clarification. In the discussion section, the relationship between trust and community resilience needs more discussion. The current discussion is somewhat scattered.

Response: Thank you for the suggestion. In the revised version of the manuscript, we have better clarified the discussion section behind the review (RL 631-661):

“Comparing the critical facets of trust delineated in the four phases of DRR, the in-itial two phases mainly exhibit trust levels influenced by the nature of received infor-mation and the credibility of its source. Conversely, in the latter two phases, trust lev-els are shaped by tangible aspects of the community, such as social infrastructure, combined with social elements like the degree of cooperation among actors (i.e., citi-zens, associations and non-governmental agencies, institutions). It appears that the evolution across the four phases of DRR entails a shift from more abstract to more concrete aspects, all the while underscoring the significance of both in the entire risk management process. However, it is important to point out that the levels of trust in the first phases may reflect the levels of trust in the last phases, indicating the need to establish a robust foundation of trust to enhance the community's overall resilience in the face of risks.

[…]

Establishing trust can significantly increase the capacity for managing disaster risks and enhancing resilience within communities. By relying on trust, individuals, com-munities, and institutions have the capacity to absorb and recover from disasters, concurrently fostering positive adaptation and transformation in their behaviors amid enduring changes and uncertainties. Consequently, there is a crucial need to promote collaborations and partnerships across various sectors, prioritizing community en-gagement and endorsing leadership models capable of expanding trust in the disaster management process. This, in turn, initiates a reinforcing cycle leading to an elevated level of trust in institutions. Moreover, the importance of sufficient information and communication is emphasized throughout the entire cycle. Research indicates that when authorities provide emergency information aligned with citizens' needs, this bolsters trust in authorities, thereby reducing the likelihood of seeking information from alternative and unreliable sources. Conversely, inadequate communication by authorities is associated with enduring negative consequences in disaster manage-ment.”

Specific comments: 

Point 1: Line 51-52, Page 2: “Consequently, trust transcends individual boundaries and maintains a close association with the concept of community.”

Please provide supporting references or evidence.

Response: Thank you for the suggestion. We have added references (RL 45-46):

“Consequently, trust transcends individual boundaries and maintains a close association with the concept of community4,8”

Point 2: Line 64-67, Page 2: “Prior research suggests that community trust is influenced by two key factors, namely residential stability and ethnic homogeneity, which are construed as products of shared meanings and associated traditions that serve to define, inform, and shape how individuals make experience of the community and their collective sense of community[9,10]”

Actually, I do not see any necessity to mention this information in the Introduction section.

Response: Thank you for the suggestion. In the revised version of the manuscript, we have modified the introduction section behind the review.

Point 3: Line 151-152, Page 4: “In fact, given the inherent challenge of assessing trust in DRR, it frequently emerges as an avenue for future research within scientific literature.”

Please provide supporting references or evidence.

Response: Thank you for the suggestion. We have added references (RL 166-167):

“In fact, given the inherent challenge of assessing trust in DRR, it frequently emerges as an avenue for future research within scientific literature3.”

Point 4: Line 173-176, Page 4: “Initially, we performed searches in academic databases (Pub-Med/Medline, ISI Web of Science, and SCOPUS), using a combination of relevant search keywords (e.g., trust, disaster risk reduction, resilience, prevention, preparedness, response, recovery, community)”

It should be “Initially, we performed searches in academic databases (Pub-Med/Medline, ISI Web of Science, and SCOPUS), using a combination of relevant search keywords (e.g., trust, disaster risk reduction, resilience, prevention, preparedness, response, recovery, community).”

Response: Thank you for the suggestion. In the revised version of the manuscript, we have corrected the punctuation.

Point 5: Line 196-198, Page 4: “Therefore, trust dynamics discussed in these articles encompass research conducted in a variety of countries across South Asia, Oceania, North and South America, and Europe, with a gradually decreasing geographical diversity.”

Is this statement(“with a gradually decreasing geographical diversity”) accurate?

 Response: Thank you for the suggestion. We corrected the expression (RL 231):

“Therefore, trust dynamics discussed in these articles encompass research conducted in a variety of countries across South Asia, Oceania, North and South America, and Europe, listed on the basis of research conducted on the topic”.

Point 6: The value and significance of this study was not clear. 

Response: Thank you for the suggestion. In the revised version of the manuscript, we have clarified the value and significance of this study (RL 713-722):

“In this article, we have critically examined the role of community and institution-al trust throughout all phases of DRR, to summarize research findings in this field and gain a deeper understanding of the connection between trust and community resili-ence in disaster risk management. Consequently, this critical review has represented an initial endeavor to consolidate and systematize the available evidence concerning the impact of trust on DRR. The findings indicate that trust plays a pivotal role in eve-ry phase of DRR, influencing the efficacy of prevention actions by governments and individual preparedness, response, and recovery from disasters. Therefore, it is imper-ative to cultivate a “culture of trust”, both at the community and institutional levels, to effectively address the risks associated with disasters.”

It is noted that the manuscript needs careful editing by someone with expertise in technical English editing paying particular attention to English grammar, spelling, and sentence structure so that the goals and results of the study are clear to the reader.

Response: Thank you for the suggestion. The revised version of the manuscript was proofread by a native English speaker who is also a professional translator.

Reviewer 2 Report

Comments and Suggestions for Authors

Thank you for the opportunity to review this very interesting review. I quite enjoyed reading it and I think it would be a good contribution to the field of DRR. Below are some suggestions to strengthen the article.

Introduction 

-      Suggest deleting the word ‘natural’ before ‘disaster’ (including your key terms), as there has been a recent shift to acknowledge that disasters are not natural but rather are underscored by political and social inequalities. See Chmutina, K., & Von Meding, J. (2019).

-      The first three paragraphs of your article do not link well. You talk about trust, then go on to talking about coping, adaptation and mitigation then move back to talking about trust. How does trust link to coping, adaptation and mitigation? 

-      Your opening line states “In recent years, there has been an increasing scholarly interest in the examination of 31 the role of trust within the domain of disaster risk reduction (DRR).” yet you do not reference any of this literature. Include some of the key literatures here. 

Methods 

-       To me it sounds like you conducted a rapid review. It might be worth reading/including Arksey & O’Malley to support your methods. 

Results 

-       suggestion to rephrase this sentence on page 5 “Even though not all disasters can be entirely averted, preventative strategies are aimed at affording long-term protection against them”. I recommend rephrasing as a lot of literature argues that all disasters can be averted. Secondly, the way this sentence is currently written, suggests that disasters are the natural hazard (earthquake, volcano, flood etc) that people need protection from. In current disaster thinking, disasters are created by policy failures and inequities. 

-       Table 2 – You note “California” under “country” for the Adams paper. 

-       Overall, the discussions presented in this section are very long and I think could benefit from shortening. What are the similarities across the papers? What are some distinct differences? Instead of describing each paper in detail, combining paper discussions together would reduce the length of this section and provide a more critical review. 

Discussion 

-       In parts, the discussion reads more like recommendations as opposed to a critical reflection of the papers reviewed. 

-       Throughout the paper, you have made the logical decision to structure your findings into the four core ‘steps’ of disaster management, however ,a suggestion is to at some point in the discussion, bring these together in the concept of trust as often, these ‘distinct’ phrases are rarely structured as such. For example, you could talk perhaps about how trust in the first phase reflects trust in the last phase and perhaps the importance of trust throughout? Understandably data is limited and the articles reviewed are situated within one phrase, however, it would be worthwhile adding your own critical reflections on what you have found in this review. 

Conclusion 

-       The conclusion reads as a limitations section and does not provide a summary of key points highlighted in the paper. 

Recommended readings: 

-       Arksey, H.; O’Malley, L. Scoping studies: Towards a methodological framework. Int. J. Soc. Res. Methodol. 2015, 8, 19–32. 

-       Chmutina, K., & Von Meding, J. (2019). A dilemma of language:“Natural disasters” in academic literature. International Journal of Disaster Risk Science10(3), 283-292.

Author Response

Response to Reviewer 2 Comments

Thank you for the opportunity to review this very interesting review. I quite enjoyed reading it and I think it would be a good contribution to the field of DRR. Below are some suggestions to strengthen the article.

Introduction

Point 1: Suggest deleting the word ‘natural’ before ‘disaster’ (including your key terms), as there has been a recent shift to acknowledge that disasters are not natural but rather are underscored by political and social inequalities. See Chmutina, K., & Von Meding, J. (2019).

Response: Thank you for the suggestion. We have better clarified the concept you suggested (RL 240-243):

“Although the literature posits that disasters frequently stem from policy failures and inequities, rendering complete avoidance plausible, preventive strategies are designed to offer sustained protection, acknowledging the potential for their future occurrence36.”

Point 2: The first three paragraphs of your article do not link well. You talk about trust, then go on to talking about coping, adaptation and mitigation then move back to talking about trust. How does trust link to coping, adaptation and mitigation?

Response: Thank you for the suggestion. In the revised version of the manuscript, we have better clarified the introduction section behind the review (RL 32-114):

In recent years, there has been an increasing scholarly interest in the examination of trust within the domain of disaster risk reduction (DRR)1–4. Researchers hailing from diverse disciplines, including economics, law, psychology, and social sciences, have collectively recognized trust as a critical factor impacting the level of community re-sponse when confronted with natural disasters.

Trust can be broadly defined as “a psychological state comprising the intention to accept vulnerability based upon positive expectations of the intentions or behavior of another”5. The concept of trust assumes a fundamental role in comprehending the dy-namics of social interactions among individuals and groups. Trust necessitates the willing acceptance of vulnerability by the trusting party when engaging with a trusted entity—whether it be an individual, a group, or an institution—without the presence of immediate guarantees or assurances regarding the motivations and behaviors of the trusted party6,7.

Consequently, trust transcends individual boundaries and maintains a close asso-ciation with the concept of community4,8. As elucidated by Walker and colleagues9, trust is essential for forming and maintaining connections between individuals who may not interact otherwise. In this context, trust can be considered as a relational con-struct that defines the quality of interpersonal communication between individuals or groups. It enables individuals to establish relationships within their communities and with institutions, and facilitates the exchange of support and assistance10. Specifically, community trust can be characterized as a fundamental attribute of communities that implies the presence of confidence in participation and the belief that a community possesses the capacity to resolve disputes and engage in collectively accepted public endeavors11.

Notably, trust exhibits the capacity to mitigate the adverse impacts of psycholog-ical distress experienced by marginalized or economically disadvantaged groups12. Conversely, numerous findings indicate that the community's vulnerability and resili-ence levels are affected by insufficient levels of trust and unsupportive governance structures, hindering access to resources and inclusion in DRR processes. This, in turn, perpetuates a self-reinforcing cycle that erodes trust13.

Furthermore, it has been observed that trust in institutions, environmental groups, and scientists strongly correlates with the adoption of appropriate behaviors in DRR. These behaviors include individual-level preparatory actions, e.g., the acquisition of insurance and the endorsement of adaptation policies8,13. For this reason, in recent years, there has also been a growing interest in the examination of trust concerning government and institutional entities. The notion of institutional trust hinges upon the belief in the capability of institutions to effectively manage a diverse array of risks and social challenges14. This concept is frequently associated with the expectation that in-stitutions will implement policies that are advantageous and successful for the well-being of citizens, and is regarded as a significant gauge of broad-based political endorsement9.

Institutional trust assumes particular significance when risks and their potentially negative impact on citizens emanate primarily from sources beyond individual con-trol, as well as when the institution is entrusted with the responsibility of forecasting and addressing damages that are largely outside the control of citizens—such as in situations involving natural hazards, where citizens cannot predict the entity of dam-age and have little control over recovery processes. The significance of trust in reduc-ing complexities and costs of risks lies in fostering a robust sense of concern, solidarity, and active engagement within the community. This, in turn, may lead to a more effi-cient response to emergencies15,16.

Institutional trust exhibits both a relational and instrumental dimension, directly tied to the outcomes of interactions between citizens and institutions. It can be affected by factors such as individual knowledge and perceived ability of emergency workers17. Accordingly, institutional trust with respect to disaster management can be bolstered through the enhancement of public knowledge as a tool within emergency prepared-ness plans4, and the perception of emergency workers’ competence (entailing confi-dence in their ability to devise effective strategies, engage with the community, and fortify community resilience) plays a pivotal role in fostering institutional trust4. Con-versely, when the processes of forecasting and mitigating damages fail to be efficiently executed, there exists the potential for the erosion of trust among disaster victims18. A decline in institutional trust signifies the belief that when adverse events occur, insti-tutions may not be relied upon to provide essential resources or to take actions aimed at ensuring safety and justice. Notably, this perception may endure over time and con-tribute to the perpetuation of psychological distress within the community19.

DRR encompasses a trio of essential measures designed to address the challenges posed by disasters, namely coping, adaptation, and mitigation20. More precisely, (a) coping denotes the capacity of individuals, organizations, and institutions to effec-tively manage risk or disaster conditions through the utilization of their available skills and resources21; (b) adaptation entails the process of making adjustments to current or anticipated risks and their corresponding impacts, with the aim of either mitigating harm or capitalizing on beneficial opportunities21; (c) mitigation encompasses the pro-cedure of reducing or minimizing the adverse consequences stemming from hazardous events 21.

Scholars concur that communities exhibiting higher levels of resilience demon-strate enhanced capacities for responding effectively to adversities, rendering them more adept at both preventing and managing disasters22. Resilience in the aftermath of a disaster is defined as the ability of a system, community, or society to resist, absorb, accommodate, adapt to, transform, and recover from the effects of a hazard in a timely and efficient manner, including the preservation and restoration of its essential basic structures and functions. It has been seen that resilience is closely related to the levels of trust23. Also, the relationship between trust and environmentally responsible behav-ior has been extensively explored in the academic literature. Trust plays a crucial role in alleviating the cognitive challenges associated with evaluating risks and making corresponding behavioral judgments throughout all phases of DRR, also improving the quality and rapidity of decision-making8. Therefore, it might be of utmost importance to critically examine the role of trust in DRR.

[…]

This makes DRR necessary for reducing the dramatic impact of disasters on communi-ties and promoting resilience. Scholars concur that communities exhibiting greater re-silience are more adept at responding effectively to various adversities22. It has been also observed that resilience is closely associated with the community's levels of trust as well23.

[…]

The current critical review focuses on the importance of trust and its role throughout all phases of disasters. It emphasizes trust as a crucial variable in DRR, investigating how trust shapes community prevention, preparedness, response, and recovery in the face of natural disasters. Previous research has sought to understand the role of trust in each distinct phase of DRR. Furthermore, various studies provide diverse definitions and interpretations of trust in DRR and various well-being outcomes. The separate examination of trust in each phase yields a complex array of outcomes, indicating the necessity to summarize and synthesize this literature. Therefore, in this critical review of literature on the significance of trust across all disaster phases, our objectives are to summarize study findings, systematize relevant knowledge, identify potential gaps in existing literature, and outline future research directions in this domain.”

Point 3: Your opening line states “In recent years, there has been an increasing scholarly interest in the examination of the role of trust within the domain of disaster risk reduction (DRR).” yet you do not reference any of this literature. Include some of the key literatures here.

Response: Thank you for the suggestion. We have added references (RL 32-33):

“In recent years, there has been an increasing scholarly interest in the examination of trust within the domain of disaster risk reduction (DRR)1–4.”

Methods

Point 1: To me it sounds like you conducted a rapid review. It might be worth reading/including Arksey & O’Malley to support your methods.

Response: Thank you for the suggestion. In the revised version of the manuscript, we have cited Arksey & O’Malley to support our methods.

Results

Point 1: suggestion to rephrase this sentence on page 5 “Even though not all disasters can be entirely averted, preventative strategies are aimed at affording long-term protection against them”. I recommend rephrasing as a lot of literature argues that all disasters can be averted. Secondly, the way this sentence is currently written, suggests that disasters are the natural hazard (earthquake, volcano, flood etc) that people need protection from. In current disaster thinking, disasters are created by policy failures and inequities.

Response: Thank you for the suggestion. We have better clarified the concept you suggested (RL 117-119/240-243):

“Nevertheless, literature on the subject underscores that disasters are not solely natural occurrences; rather, they are shaped by policy failures and inequities25.

[…]

Although the literature posits that disasters frequently stem from policy failures and inequities, rendering complete avoidance plausible, preventive strategies are designed to offer sustained protection, acknowledging the potential for their future occurrence36.”

Point 2: Table 2 – You note “California” under “country” for the Adams paper.

Response: Thank you for the suggestion. In the revised version of the manuscript, we have corrected the country.

Point 3:  Overall, the discussions presented in this section are very long and I think could benefit from shortening. What are the similarities across the papers? What are some distinct differences? Instead of describing each paper in detail, combining paper discussions together would reduce the length of this section and provide a more critical review. 

Response: Thank you for the suggestion. In the revised version of the manuscript, we have better clarified the discussion section behind the review (RL 631-641):

“Comparing the critical facets of trust delineated in the four phases of DRR, the in-itial two phases mainly exhibit trust levels influenced by the nature of received infor-mation and the credibility of its source. Conversely, in the latter two phases, trust lev-els are shaped by tangible aspects of the community, such as social infrastructure, combined with social elements like the degree of cooperation among actors (i.e., citi-zens, associations and non-governmental agencies, institutions). It appears that the evolution across the four phases of DRR entails a shift from more abstract to more concrete aspects, all the while underscoring the significance of both in the entire risk management process. However, it is important to point out that the levels of trust in the first phases may reflect the levels of trust in the last phases, indicating the need to establish a robust foundation of trust to enhance the community's overall resilience in the face of risks.”

Discussion

Point 1: In parts, the discussion reads more like recommendations as opposed to a critical reflection of the papers reviewed.

Response: Thank you for the suggestion. In the revised version of the manuscript, we have better clarified the discussion section behind the review

Point 2: Throughout the paper, you have made the logical decision to structure your findings into the four core ‘steps’ of disaster management, however, a suggestion is to at some point in the discussion, bring these together in the concept of trust as often, these ‘distinct’ phrases are rarely structured as such. For example, you could talk perhaps about how trust in the first phase reflects trust in the last phase and perhaps the importance of trust throughout? Understandably data is limited and the articles reviewed are situated within one phrase, however, it would be worthwhile adding your own critical reflections on what you have found in this review.

Response: Thank you for the suggestion. In the revised version of the manuscript, we have better clarified the discussion section behind the review (RL 647-661):

“Establishing trust can significantly increase the capacity for managing disaster risks and enhancing resilience within communities. By relying on trust, individuals, com-munities, and institutions have the capacity to absorb and recover from disasters, concurrently fostering positive adaptation and transformation in their behaviors amid enduring changes and uncertainties. Consequently, there is a crucial need to promote collaborations and partnerships across various sectors, prioritizing community en-gagement and endorsing leadership models capable of expanding trust in the disaster management process. This, in turn, initiates a reinforcing cycle leading to an elevated level of trust in institutions. Moreover, the importance of sufficient information and communication is emphasized throughout the entire cycle. Research indicates that when authorities provide emergency information aligned with citizens' needs, this bolsters trust in authorities, thereby reducing the likelihood of seeking information from alternative and unreliable sources. Conversely, inadequate communication by authorities is associated with enduring negative consequences in disaster manage-ment.”

Conclusion

Point 1: The conclusion reads as a limitations section and does not provide a summary of key points highlighted in the paper.

Response: Thank you for the suggestion. In the revised version of the manuscript, we have better clarified the conclusion section behind the review (RL 713-722):

“In this article, we have critically examined the role of community and institution-al trust throughout all phases of DRR, to summarize research findings in this field and gain a deeper understanding of the connection between trust and community resili-ence in disaster risk management. Consequently, this critical review has represented an initial endeavor to consolidate and systematize the available evidence concerning the impact of trust on DRR. The findings indicate that trust plays a pivotal role in eve-ry phase of DRR, influencing the efficacy of prevention actions by governments and individual preparedness, response, and recovery from disasters. Therefore, it is imper-ative to cultivate a “culture of trust”, both at the community and institutional levels, to effectively address the risks associated with disasters.”

Recommended readings: 

 -       Arksey, H.; O’Malley, L. Scoping studies: Towards a methodological framework. Int. J. Soc. Res. Methodol. 2015, 8, 19–32. 

-       Chmutina, K., & Von Meding, J. (2019). A dilemma of language:“Natural disasters” in academic literature. International Journal of Disaster Risk Science, 10(3), 283-292.

Response: Thank you for the suggestion. In the revised version of the manuscript, we have integrated these references.

Reviewer 3 Report

Comments and Suggestions for Authors

Dear Author,

Many thanks for the opportunity to review your fine paper. For your consideration I would suggest that the limitations are separated out of the conclusions into its own section. I would also suggest that the methods are more accurately described (and the limitations of the methods articulated, particularly as this is a literature review I think it is reasonable to assume that without robust search strategy (i.e. systematic) that some research may not have been identified).

Overall a useful contribution, I would have thought ethics and decision making may have been captured.

Author Response

Response to Reviewer 3 Comments

Dear Author,

Many thanks for the opportunity to review your fine paper. For your consideration I would suggest that the limitations are separated out of the conclusions into its own section. I would also suggest that the methods are more accurately described (and the limitations of the methods articulated, particularly as this is a literature review I think it is reasonable to assume that without robust search strategy (i.e. systematic) that some research may not have been identified).

Response: Thank you for the suggestion. In the revised version of the manuscript, we have more accurately described the methods and we have separated limitations out of the conclusions.

Overall a useful contribution, I would have thought ethics and decision making may have been captured.

Response: Thank you for the suggestion. In the revised version of the manuscript, we have better clarified the ethics and decision making strategies in the discussion section behind the review (RL 689-711):

“Through proficient use of communication channels (e.g., mass media or social media), experts in disaster management can facilitate the dissemination of emergency infor-mation and the cultivation of trust. Consequently, trust contributes to augmenting community awareness and fostering a more profound comprehension of hazard pro-cesses, playing a pivotal role in bolstering community resilience67. In the realm of dis-aster management, institutional policies should strive to enhance individuals' trust in their community (e.g., relatives, neighbors, coworkers) and institutions (e.g., agencies, authorities, government). The establishment of trust holds the potential to significantly fortify the capacity for disaster risk prevention, management, and resilience in com-munities.

Therefore, the findings reported in this review facilitated the comprehension re-garding how the level of trust is intrinsically linked to a community's resilience. Spe-cifically, studies have indicated that trust serves as a direct predictor of resilience44,56. Similar to trust, resilience to disasters extends across all phases of a disaster (pre-event, during a disaster, and post-event phases) and is not confined solely to the disaster re-sponse and recovery stages74. Resilience is thus contingent on the community's capaci-ty to recover post-disaster and is also connected to the extent of resources and capabil-ities the community possesses, both before and after a disaster. An integral dimension of resilience involves fortifying relationships among citizens, NGOs, and government agencies, which are indispensable for establishing a robust foundation of trust and co-hesion within the community and its collaborative partners when faced with disasters. Consequently, resilience and trust appear to be mutually influential and interdepend-ent during disasters.”